# Position: the Stochastic Parrot in the Coal Mine
# Model Collapse is a Threat to Low-Resource Communities

**Devon Jarvis** [* 1 2]  **Richard Klein** [1 2]  **Benjamin Rosman** [2 1]  **Steven James** [1 2]  **Stefano Sarao Mannelli** [3 1]

## Abstract

Model collapse, the degradation in performance that arises when generative models are trained on the outputs of prior models, is an increasing concern as artificially generated content proliferates. Related critiques of large language models have highlighted their tendency to reproduce frequent patterns in training data, their reliance on vast datasets, and their substantial environmental cost. Together, these factors contribute to data degradation, the reinforcement of cultural biases, and inefficient resource use. In this position paper we aim to combine these views and argue that model collapse threatens current efforts to democratize AI. By reducing training efficiency and skewing data distributions away from the tails of their support, model collapse disproportionately impacts low-resource and marginalized communities. We examine both the environmental and cultural implications of this phenomenon, situate our position within recent position papers on model collapse, and conclude with a call to action. Finally, we outline initial directions for mitigating these effects.

## 1. Introduction

The emergence of generative artificial intelligence (AI) techniques, such as transformers (Vaswani et al., 2017) and diffusion models (Rombach et al., 2022), has led to widespread public adoption in recent years. ChatGPT alone is estimated to generate approximately $0.1\%$ of all words produced globally each day (Schaeffer et al., 2025). This adoption is already reshaping how people work and learn (Chen et al.,

2020; Brynjolfsson et al., 2025). While generative AI has the potential to be broadly democratising, the consequences of its large-scale use remain poorly understood (Kosmyna et al., 2025; Stankovic et al., 2025).

The low cost of content generation has led to an unprecedented growth in available data (Wei & Tyson, 2024), with recent estimates suggesting that generated content constitutes roughly 30% of the internet as of 2025 (Spennemann, 2025). As a result, successive generations of generative models are trained on increasing amounts of AI-generated data. Several recent studies show that training repeatedly on such "synthetic" data degrades generation quality across iterations (Zhu et al., 2025; Shumailov et al., 2024; Dohmatob et al., 2024a; Seddik et al., 2024). This phenomenon, known as model collapse, leads to highly repetitive or incoherent text and overly stylized, unrecognisable images (Bohacek & Farid, 2023; Dohmatob et al., 2024c; Gerstgrasser et al., 2024).

Parallel work has also raised concerns around the propensity of language models to reinforce stereotypes and biases present in their training data, and the significant computational resources required to train them (Bender et al., 2021; Samsi et al., 2023; Jegham et al., 2025). Due to the expense in collecting data from communities less prevalent on the internet (Kandpal & Raffel, 2025) and training such extremely large models, low-resource communities have less representation in emerging technologies which hinders the applicability of the technology to their contexts.

At the same time, the energy demands of large-scale computation continue to drive new infrastructure development in already energy-intensive regions (Electricity, 2024), while the environmental consequences are borne globally. Low-resource communities are particularly vulnerable to these effects, including increased exposure to climate-related disasters (Van Aalst, 2006). As a result, these communities bear a disproportionate share of the costs while receiving comparatively little of the benefit (Farnadi et al., 2024). As Strubell et al. (2019) note, marginalized communities that experience the effects of environmental change first are often the last to gain access to new technologies.

---

[1]School of Computer Science and Applied Mathematics, University of the Witwatersrand, Johannesburg, South Africa [2]Machine Intelligence and Neural Discovery Institute, University of the Witwatersrand, Johannesburg, South Africa [3]Data Science and AI, Computer Science and Engineering, Chalmers University of Technology and University of Gothenburg. Correspondence to: Devon Jarvis <devon.jarvis@wits.ac.za>.

*Proceedings of the $43^{rd}$ International Conference on Machine Learning*, Seoul, South Korea. PMLR 306, 2026. Copyright 2026 by the author(s).

In the context of language models, these structural imbalances are compounded by the phenomenon described by Bender et al. (2021) as "Stochastic Parrots". While large language models can produce fluent and coherent text when trained on vast datasets, their outputs remain ungrounded, lacking social context and adherence to human norms. The consequence is that the models repeat statements based on frequency rather than merit or accuracy (Orgad et al., 2025). Thus, generated text can be skewed by social constructs that are potentially harmful and factually incorrect. Even in the case of hallucinations, the errors are not arbitrary and reflect an underlying bias of the models against under-represented groups (Farnadi et al., 2024; Orgad et al., 2025). Coupled with the human proficiency at finding meaning as an inter-locutor (Clark & Bangerter, 2004; Clark & Krych, 2004; Janzen & Shaffer, 2008), this creates systems that can be misleading or harmful at scale (Choi et al., 2024). Although we revisit the continued relevance of the Stochastic Parrot analogy after several years of progress in Sec.4, it remains foundational to the concerns raised in this perspective.

While model collapse, the energy consumption of Large Language Models (LLMs) and their propensity to repeat hegemonic or biased viewpoints are pressing issues in isolation, their interaction has received little attention. We begin to close this gap by arguing that **model collapse amplifies the existing harms of LLMs for low-resource languages and marginalized communities by producing models which favour hegemonic viewpoints more with each generation, remove under-represented viewpoints from the data distribution and scale poorly with the addition of real data. By poisoning the data and removing the tails of the distribution, it undermines ongoing efforts to democratize AI. Addressing model collapse should therefore be a central concern for research on low-resource settings and AI fairness.**

To substantiate this claim, we begin in Sec.2 by reviewing the recent theoretical and empirical findings on model collapse. In Sec.3 we reconsider the points raised in the seminal work by Bender et al. (2021) on Stochastic Parrots in light of the new findings on model collapse (Dohmatob et al., 2024c; Shumailov et al., 2024). We highlight how model collapse exacerbates both the environmental and social risks associated with large language models, while also introducing new risks. Sec.4 provides some alternative views which might mitigate the potential risks of model collapse on low resource settings. Sec.5 concludes with a call to action and initial recommendations for immediate directions of research to better understand or mitigate the risks of model collapse for low-resource and marginalized communities.

Throughout this work we make use of the term "low-resource community" and it is important to specify what we mean. In this case, the term is not limited to reflecting the fact that the community has a relatively smaller proportion of their data present in datasets or on the internet, although this is part of it. Through this term we also include communities which lack sufficient compute or financial means to train large models or collect data freely, relative to the dominant communities on the internet. Furthermore, this notion of a dominant community is highly context specific and temporal. A community may be dominant is a particular dataset and not within another. Importantly, there are a number of reasons why a community's data may occupy the tails of a data distribution and we do not consider any one reason here. We do, however point to a few beyond a lack of representation throughout this work. Our point here is not to group all of these concerns into one, but to argue that these reasons and difficulties exist disproportionately for some communities which may lead to more severe model collapse for these same communities. Thus, the term "low-resource community" is used for ease of communication and not intended to convey a uniform group. Understanding how model collapse manifests for different reasons is something we touch on in our call-to-action in Sec.5. Additionally, Tab.1 summarizes the main mechanisms of model collapse discussed throughout the paper, linking each cause to its expected effect on low-resource communities.

**Conflict of Interest Disclosure**: The authors have no conflict of interest to disclose.

## 2. Background

While the precise definition of model collapse is still evolving and often conflates distinct effects (Schaeffer et al., 2025), the literature generally groups several phenomena under this heading, ranging from a catastrophic divergence in test loss – often an artefact of unrealistic "replace" training paradigms (Schaeffer et al., 2025) – to subtler deformations of the data distribution. Despite this definitional ambiguity, the core concern generally refers to a specific failure mode: *a progressive decrease in the marginal value of additional training data (when a proportion of that data is synthetic) and a simultaneous reduction in the diversity and quality of the output distribution* (Shumailov et al., 2024).

This degradation is best understood through the lens of neural scaling laws that describe the predictable power-law improvement in model performance (usually measured by cross-entropy loss) as compute, dataset size, and parameter count increase (Kaplan et al., 2020; Bahri et al., 2024). These laws provide an optimistic alternative to the performance plateaus typical of overfitting in earlier architectures (Cogswell et al., 2015; Santos & Papa, 2022). However, the introduction of synthetic data fundamentally alters this trajectory. Theoretical and empirical works indicate that at sufficient scales, even a small fraction of synthetic data (around 1%) causes models to suffer from a change in scal-

| Cause / mechanism | Why it matters for low-resource communities | Likely effect | Concerns |
|---|---|---|---|
| Synthetic-data feedback loops (Whitney & Norman, 2024) | Small real-data distributions are more easily overwhelmed by generated content. | Dominant patterns are reinforced across generations, while rare linguistic and cultural forms are pushed further into the tail. | Representation loss |
| Tail erasure under model collapse (Wyllie et al., 2024) | Low-resource languages, dialects, and marginalized viewpoints often already occupy low-probability regions of the data distribution. | Loss of diversity, reduced expressivity, and weaker model performance on communities whose data is already scarce. | Cultural erosion |
| Changed scaling behaviour with synthetic data (Shumailov et al., 2024) | If more data yields less improvement, communities with limited access to real data and compute have fewer viable routes to improvement. | Higher data and compute requirements, widening the gap between high- and low-resource settings. | Unequal access |
| High cost of real data acquisition (Kandpal & Raffel, 2025) | Collecting, validating, and maintaining culturally representative data is expensive and often unpaid or under-resourced. | Low-resource communities may be pushed toward synthetic supplementation, increasing the risk of poisoning their own data. | Data inequity |
| Automated curation and synthetic-data detection (Boutadjine et al., 2025) | Synthetic detection and filtering systems may be unreliable, opaque, or biased against uncommon language use. | Authentic but rare or reclaimed language may be removed, while synthetic or hegemonic content remains. | Curation bias |
| Value-lock and hegemonic viewpoints (Bender et al., 2021) | LLMs tend to reproduce frequent historical patterns rather than socially evolving or locally grounded meanings. | Outdated or dominant viewpoints persist, while new terms, norms, and movements struggle to enter the model distribution. | Bias reintroduction |
| Multilingual transfer and language drift (Lu et al., 2020) | Low-resource languages often depend on shared multilingual representations learned from high-resource languages. | Fine-tuning or repeated generation may erode low-resource capabilities faster than high-resource ones. | Language drift |
| Tokenizer and morphology mismatch (Rajab, 2022) | Agglutinative languages, code-switching, and non-standard orthographies may create longer-tailed token distributions. | Model collapse may manifest differently across languages, making universal mitigation strategies unreliable. | Measurement gap |

*Table 1.* Mechanisms by which model collapse may disproportionately affect low-resource and marginalized communities.

ing exponents – essentially hitting a "glass ceiling" where larger datasets no longer yield proportional improvements (Dohmatob et al., 2024b;a; Cui et al., 2025). This phenomenon is explained as the model overfitting to the dominant factors of variation in the original dataset which are then over-represented in the generated samples (Shumailov et al., 2024). The generated samples enter the training data for subsequent models, strengthening these dominant factors further and creating a feedback loop where the model progressively loses the ability to represent the variability and structure of real-world data (Shumailov et al., 2023; Bohacek & Farid, 2023).

This over-representation of dominant factors leads directly to a loss of variability through this feedback loop that systematically erases rare or "tail" examples (Shumailov et al., 2023; Bohacek & Farid, 2023). Consequently, the rich variance of the original real-world distribution is replaced by a smoothed, homogenized approximation. The threat of model collapse is, therefore, not necessarily that models will become gibberish, but that they will become narrow—capable of reproducing frequent patterns but increasingly blind to the long tail of human diversity (Schaeffer et al., 2025; Shumailov et al., 2024).

This process is closely related to a similar concept in cognitive science known as iterated learning (IL) which has been used to explain the refinement of language and behaviours over generations of human learners (Smith et al., 2003). In humans, the result is a highly structured (termed regular) language or set of actions that are easy to learn and can be composed to generalize systematically (Kirby et al., 2008). However, in both empirical experiments and theoretical models, expressivity can be lost without external intervention and the differences between closely related data points is forgotten (Kirby et al., 2008; 2014; Jarvis et al., 2026). Thus, to consider the inductive bias of the model to focus on dominant factors of variation as "only negative" is not grounded in the broader literature on cognitive modelling and machine learning. It is plausible that the real-world situation is more optimistic than those shown on smaller (but still naturalistic) datasets and in theoretical models. This is a point we return to in Sec.4. However, when coupled with the existing biases of LLMs, such as promoting the hegemonic viewpoints in its training dataset (Bender et al., 2021), and the already enormous costs involved in training them (Xia et al., 2024; Kandpal & Raffel, 2025), it is important to consider the potential harms which model collapse may cause for low-resource and marginalized communities.

# 3. Generations of Stochastic Parrots

## 3.1. Environmental and Financial Costs

The financial costs associated with training LLMs are often scrutinized (Strubell et al., 2019); for example, training a single BERT base model without hyperparameter tuning on GPUs was estimated to consume as much energy as a flight across America (Strubell et al., 2019), training a large transformer model with neural architecture search emits as much $CO_2$ emissions as the average human over 60 years (Vaswani et al., 2017; Bender et al., 2021), and training GPT-4 reportedly cost over \$100m (O'Donnell & Crownhart, 2025). However, this estimate does not consider the cost of the data used to train the models (often because it goes unpaid). Based on conservative estimated of wage rates, the costs of obtaining the training datasets are 10–1000 times larger than the costs to train the models themselves (Kandpal & Raffel, 2025). Even if the real cost is highly discounted, there is still an opportunity cost and loss of human capital associated with the use of such expensive data.

In many cases, the high cost of training an LLM is justified by its use as a foundation model (Xu et al., 2024). Hence, the cost is amortized over its entire life-cycle and uses (Ren et al., 2024). However, most research establishing the costs of training LLMs is based (necessarily) in idealized settings and with clean datasets (Zeighami et al., 2025). For example, Strubell et al. (2019) estimate that an increase in BLEU score of 0.1 from using neural architecture search for English to German translation results in an increase of \$150,000 in compute costs. Further, this cost profile shifts drastically with new modalities: generating a five-second video can consume more than 700 times the energy of generating an image (O'Donnell & Crownhart, 2025). Consequently, the compute cost of training the largest deep learning models increased roughly 300,000-fold between 2015 and 2021 (Devlin et al., 2019), a trend expected to accelerate with large-scale infrastructure projects such as the \$500 billion "Stargate" initiative (O'Donnell & Crownhart, 2025).

Using the linear model of model collapse proposed by Dohmatob et al. (2024b), achieving a tenfold reduction in test loss (from $10^{-1}$ to $10^{-2}$) requires an order-of-magnitude increase in dataset size (from $10^3$ to $10^4$), consistent with established power-law scaling (Kaplan et al., 2020). Considering a dataset where $10\%$ is synthetic, the increase in dataset size needed is *another* order of magnitude (from $10^3$ to $10^5$). While it is important to recognize that this is a theoretical model (including a hyperparameter to control the quality of the synthetic data), even if the model is pessimistic, model collapse can only stand to increase the amount of data and compute needed to train new models. Coupled with the point that the hidden costs of producing the training data are often overlooked (Kandpal & Raffel, 2025), this increase in cost becomes even more severe.

While the natural answer of collecting more real data may be viable in some settings, if the cost of obtaining data is even on the lower end of the estimate by Kandpal & Raffel (2025), then this will likely not be a viable solution for many populations. This leaves low-resource communities unable to obtain data without significant cost or without the use of synthetic data as a mechanism to supplement datasets. Thus, when benchmarking the environmental cost of training foundation models it is important to also consider the effect of cheap synthetic data or expense of real data on this process, or else it stands to be highly optimistic. This inequity is compounded by the fact that data centres are often clustered in regions with carbon-intensive grids, resulting in emissions 48% higher than the national average (O'Donnell & Crownhart, 2025).

While it is unrealistic to expect uniform technological advancement across different communities and sub-fields, the typical belief is that progress in one space will lend insight and lead to advancement in another. Instead, model collapse may result in fundamentally uneven access to new technology between communities. Moreover, the communities most hindered are precisely those that could benefit the most from technological advancement. Even under more optimistic scenarios, model collapse threatens to undermine ongoing efforts to reduce the environmental impact of training LLMs (Huang et al.; Iftikhar et al., 2025; Shi et al., 2025) and the injustice this causes (Strubell et al., 2019). Thus, beyond unequal access to new technology and the disproportionate impact of environmental harm on certain sub-populations, we hypothesize that model collapse and the cost of data acquisition could introduce a third disadvantage to low-resource communities: *a lower upper-bound on technological advancement* from these models.

## 3.2. Cultural Costs

The source of injustice from LLMs is not only in the resources required to train and use the models, however. There are also valid concerns stemming from the effect of LLMs on public perception and culture. Foundational to this section is the recognition that model collapse results in a lack of diversity of generated content and that what remains in the data is the highly probable content from the original dataset (Alemohammad et al., 2023; Bertrand et al., 2024). Thus, data which would occupy the tails of a probability distribution are forgotten or corrupted (Shumailov et al., 2024). Datasets used to train generative models are typically scraped from the internet, which has been shown to be problematic (Basta et al., 2019; Birhane & Prabhu, 2021; De Vries et al., 2019; Kurita et al., 2019). This has resulted in the models encoding biases and derogatory associations in some cases (Bao et al., 2024; Shin et al., 2024). Even when there is not overt harm, the models favour the hegemonic viewpoint presented on the internet sites where the data is

scraped from (Bender et al., 2021) – one of the foundational points leading to the stochastic parrot phrasing.

The comprehension of LLMs regarding minority groups has often been shown to be limited, absent, or stereotyped. Farnadi et al. (2024) comprehensively analyse the types of discrimination LLMs perform, identifying not only privacy issues — as LLMs tend to memorize under-represented data more — but also "hallucination gaps" where models hallucinate more facts regarding minority groups. Furthermore, they note "underspecification disparities" where models generate seemingly arbitrary text when addressing topics related to marginalized groups (Farnadi et al., 2024). Consequently, LLMs flatten complex concepts (Gallegos et al., 2024), often aligning towards Western and English-speaking values (Rao et al., 2023). Together, these factors produce an outcome that amplifies bias and removes diversity. This effect is further amplified by standard compression and acceleration techniques — including knowledge distillation, quantization, pruning, and caching — which have been shown to further increase bias (Silva et al., 2021; Ahn et al., 2022; Kirsten et al., 2025). Cases of biased data generation are also not limited to LLMs. For example, it has been shown that stable diffusion has a propensity to perpetuate stereotypes regarding race and skin tone which has a homogenising effect on less frequently represented racial and ethnic groups (Wilson et al., 2025).

Data curation has been proposed as a potential solution to this problem (Birhane & Prabhu, 2021; Jo & Gebru, 2020), but there are a number of ways that model collapse can undermine the curation process. If mitigating model collapse becomes a part of data curation, then it will be necessary to detect synthetic data for its removal. However, this is becoming an extremely difficult task as generative models become more convincing to human perception (Boutadjine et al., 2025). Thus, curation will rely more on automated detection systems. Although there has been some success in automating data mixing and curation through methods like instruction-tuning (Jiang et al., 2024; Raj et al., 2024; Liu et al., 2025; Thudi et al.) these approaches are still potentially fallible, uninterpretable and require large quantities of real or synthetic data to train (Chaka, 2023; Elkhatat et al., 2023; Ghiurău & Popescu, 2024). This could stand to worsen the of curation to remove data from marginalized groups or those who express fair but less frequent views. For example, the Colossal Clean Crawled Corpus (Raffel et al., 2020) was cleaned by discarding any sample with one of 400 bad words. While this is effective at removing some of the worst known biases and slurs from data, it also reverses any attempts to reclaim works by marginalised groups (Bender et al., 2021). This effect of LLMs to re-establish outdated connotations for words is termed "value-lock" (Bender et al., 2021) and disturbs the role of language in shaping social norms and culture. As Twyman et al. (2017) note, a central aspect of recent social movements revolves around using online forums to document events, promote coverage of new events and marginalized perspectives, and change the perspectives on existing knowledge in light of new information.

Value-lock was already a concern for first-generation LLMs based only on generated content being more likely to represent previously dominant (and hence frequently represented in the training data) viewpoints. The likely outcome of model collapse is that these dominant viewpoints will become even more over-represented in the dataset as subsequent generations of models are trained (Whitney & Norman, 2024). Equally, the less frequently encountered language which is deliberately used to change social perception will be forgotten over the generations. Wyllie et al. (2024) contextualize this within the rapid increase of LLM-generated text, defining the *"Fairness Feedback Loop."* They argue that model collapse is not merely a quality issue; it is an erasure engine that systematically removes under-represented communities from the digital reality defined by AI models. All of this stands to reinforce existing regimes of power and undermine attempts to promote social change.

Importantly, the looming harm from model collapse is not limited to a shift of the data distributions towards previous views, as shown by recent cases in high-resource settings. For example, Wang et al. (2025) show that GPT-2 favours right-leaning media when successively fine-tuned on a balanced media dataset from the US such that it undergoes model collapse. *Thus, even in ideal settings model collapse has the propensity to inject bias where there seemingly is none, or based on underlying biases which are not immediately obvious to the expert and public raters of their dataset.*

### 3.3. Transfer Learning

Recent work on low-resource languages has increasingly relied on transfer learning from multilingual models (Lai et al., 2023; Devlin et al., 2019). The rationale is that combining data from multiple languages—particularly high-resource ones—provides sufficient data to learn a semantically rich embedding space. The low-resource language can then leverage this space, enabling the model to represent meaning more effectively despite limited direct data. However, fine-tuning in this manner introduces several issues, including language drift, where the embedding space is progressively forgotten as the model is trained on a more narrowly defined task (Lee et al., 2019; Lu et al., 2020). In other words, while transfer learning provides benefits, the model can lose capabilities originally acquired during pretraining on a rich embedding space.

How model collapse manifests in this setting where only data at the edge of the distribution (the low-resource language) is being generated is an open question. If the model remains multi-lingual, then model collapse will likely result

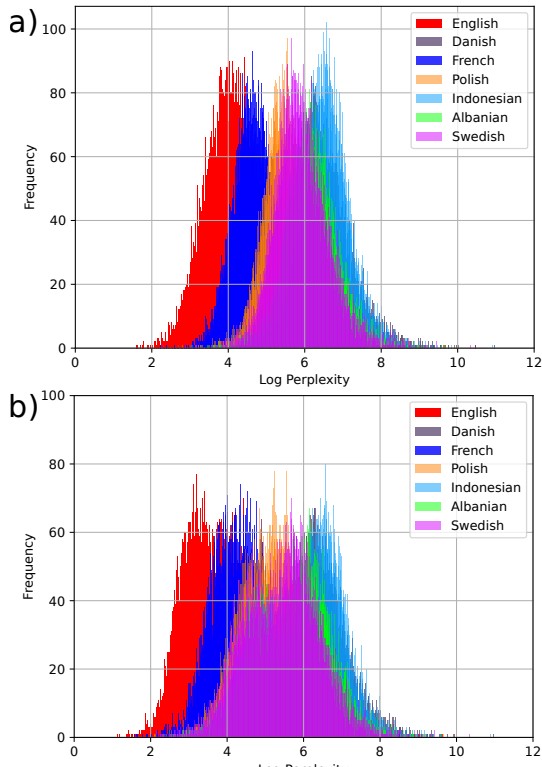

*Figure 1.* Perplexity of multiple languages using the Latin alphabet (potentially with some added characters) calculated using a pretrained GPT-2 (Radford et al., 2019). We present this result to support our position only.
a) **Original Distribution:** Note how the lower-resource languages occupy a distribution closer to the tails (at a higher perplexity) than the more high-resource languages such as English. Each language distribution is calculated using 15000 input sentences (agentlans, 2025) and show log-perplexity for visibility.
b) **Collapsing Distribution:** Note how the low-resource languages shift the most when 5000 samples are generated for each language and added to the distribution with replacement. Swedish and Polish are affected worse than the other languages, while all distributions shift left and begin to overlap – highlighting the homogenising effect of collapse.

in the erosion of low-resource languages from the data over time, and the model will progressively deteriorate faster on these languages. Figure 1 demonstrates this by showing the distribution of log-perplexity values GPT-2 (Radford et al., 2019) attributes to 16000 sentences from various languages using the Latin alphabet (agentlans, 2025). Clearly, high-resource languages such as English and French have a lower initial perplexity compared to Indonesian and Albanian. By our current understanding of model collapse, these languages at the tails of the combined distribution should be forgotten first, which is supported by the initial shift in distributions when just 2000 generated samples are added. While this is a preliminary, it is still a stark result demonstrating more need for work focusing on the effect of model collapse on low-resource languages.

Multi-lingual models are also often told which language to

generate text for (or at least prompted in a particular language) and so it is unlikely to collapse in the same manner as if it was generating text from its intrinsic distribution of language. This deliberate mechanism introduced by the human user is a means by which model collapse could be mitigated. However, prompting and guiding the generation in this manner reintroduces a mechanism for bias (Yang et al., 2023; Hida et al., 2025; Chisca et al., 2024). These and many more questions, such as the effect of model collapse on languages where code-switching is common (Woolford, 1983) (and itself a lingering issue for many low-resource languages (Yoo et al., 2025a;b; Mohamed et al., 2025)) remain open and are important to fully grasp the effect model collapse could have on marginalized and low-resource communities.

This leads to our final consideration: the deliberate use of generative models to produce content for low-resource settings and marginalized communities. One reason this is needed is to ensure that this data remains represented in the data distributions of multi-lingual models to some degree. In essence, having synthetic data for your community enter into the broader distribution is likely better than leaving your community's data to slide further into the tails. Implicitly, there is a race between communities to keep pace with the rate of data generation of the others. Real data is certainly the best content to produce, but this favours communities with resources, as noted in Sec.3.1. The net result is that marginalized communities may be pushed to poison their data merely to ensure their data remains salient in combined datasets. The other reason marginalized communities may still need to use generative models, and likely the larger reason, is to keep pace with the growing productivity internationally which can result from new technologies. Thus, these communities will need to adopt generative models to remain economically active within a globalized society with blurred cultural boundaries.

The big question is then: how quickly will content be generated in these settings with AI, compared to human content creation? In the best case, content generation will follow the same distribution of general content on the internet. In this case, all communities will have roughly the same percentage of synthetic content in their training data at a given time and will face a similar degradation from model collapse. Since the plateau from model collapse gets worse as more data is added (Shumailov et al., 2024), this may even make model collapse initially worse for high-resource settings. However, it seems plausible that the technology will provide a proportionally larger effect on low-resource settings as the cost of generating content becomes significantly cheaper. Essentially, synthetic content generation is democratised far quicker than the internet as a whole. In this case the rate of generated content production will more closely follow the population of a country, while the rate of real content generation will align to resource access. Thus, in absolute terms,

the amount of content generated by low-resource communities will be less but the proportion of synthetic data in these languages will be greater. *This will result in low-resource communities encountering severe model collapse far quicker than high-resource communities and the erosion of these cultures, before we have the methods to reverse the collapse.*

## 4. Alternative Views

There are a number of alternative views to be considered, ranging from the problem being real but mitigated, to model collapse (or at least the bottleneck imposed by generations of trainers) being beneficial. Considering the former, the extent of model collapse is still being debated and some work does show that models can stabilize after a certain number of generations (Kazdan et al., 2025). Similarly, cases where extreme collapse are shown tend to be on rather small datasets and measured using metrics like the perplexity of individual samples (Shumailov et al., 2023). While these metrics and experiments match qualitative evaluations of the generated content, how to accurately measure or characterize model collapse remains a valuable question. Due to this difficulty, isolating its effects is also difficult. A subtle but important alternative view may consider whether model collapse is really accelerating or introducing bias in the data distributions or just a symptom of a growing gap between communities based on access to resources. Thus, it is plausible that with sufficient real data, models will not be overwhelmed by synthetic data and that the data distributions may stabilize to content which is no more biased than it otherwise would be in the first generation (Alemohammad et al., 2023; Kazdan et al., 2025).

The comparison of model collapse to iterated learning supports the latter view that model collapse may even be beneficial. As noted in Sec.2, the generational transmission of language and the regularity this affords is a benefit for human language use and cognition (Kirby et al., 2008). One could argue that model collapse will result in the loss of invalid information (which includes biases) due to the fact that it is less grounded within the environment.[1] As a result, the language which will be most represented will be grounded language which expresses truths about nature. In a sense, there are many ways to be wrong and so no one way should dominate the data distribution.

It is important to note that the iterate learning algorithm can also result in a loss of expressivity when run with humans and some artificial agents (Kirby et al., 2008; 2014; Jarvis et al., 2026). What is required is a mechanism for the reintroduction of words and forgotten information and some creativity from the speakers. It is conceivable that with such

---

[1]We certainly run the risk of oversimplification here, as the degree to which biases are ingrained into the environment once they exist is a subtle point beyond the scope of this discussion.

a mechanism, LLMs will contribute to the natural evolution of language without causing collapse. This process could even be partially achieved in the near-term through prompting and curriculum learning (Jiang et al., 2024), the inclusion of richer multi-modal context windows, and frameworks such as algorithmic reparation (Davis et al., 2021; Wyllie et al., 2024). Once again, this will require more data and deliberate use by different communities to fight model collapse. Moreover, the worse the collapse, the more difficult it will be to reintroduce sufficient language to stabilise the data distribution. If any communities have a smaller margin for error and should be seeking such interventions, it is the low-resource and marginalized communities.

Schaeffer et al. (2025) presents one important prior position paper which aimed to provide some needed precision in the discussion of model collapse. The primary point of Schaeffer et al. (2025) is that the literature on model collapse both conflates multiple definitions and over-simplifies the phenomenon. One of the primary critiques is that within a given paper, the definition of model collapse changes and distinct phenomena are used to characterize its occurrence. We fall into the same trap here: Sec.3.1 relies on a notion of model collapse that causes a change in neural scaling laws (Dohmatob et al., 2024c, and Definition 5 in Schaeffer et al. (2025)), while Sec.3.2 focuses mainly on the loss of data at the tails of the original distribution (Shumailov et al., 2023; Wyllie et al., 2024; Shumailov et al., 2024, and Definition 7 in Schaeffer et al. (2025)). It is not our aim here to argue for a particular definition, but rather to show that regardless of the definition used it is plausible that model collapse will disproportionately affect low-resource communities.

From the three "types" of model collapse definitions identified by Schaeffer et al. (2025), we have mainly considered the consequences of two types. Specifically, we show that type 2, which considers the deformation of the data distribution, will likely first erase marginalized community data and promote the hegemonic viewpoints more so than even prior works considered (Bender et al., 2021). Secondly, we considered type 3, which is the decreasing value from additional data, and reconsidered the already alarming trend of marginalized communities disproportionately suffering the costs of environmental damage while benefiting the least from the new technologies (Strubell et al., 2019). We extend this with the recent point from Kandpal & Raffel (2025) claiming that data acquisition is the most expensive part of a training pipeline, and highlight that strategies revolving around collecting more data to mitigate model collapse will only be feasible for communities able to "pay" this data cost.

The type 1 definition of Schaeffer et al. (2025) is the progressive inability of the model to fit the original data distribution due to the introduction of synthetic data over generations. This type of definition is certainly important, particularly

when lending theoretical understanding. However, this effect will also hit low-resource communities first as these data distributions will be more easily overwhelmed by synthetic data. Secondly, this type of definition provides a slightly more long-term perspective, while we are arguing for the immediate consideration of model collapse by low-resource communities. It is also worth noting that the intermediate generations are where the cultural harms noted in Sec.3.2 will arise, long before the generated content becomes incomprehensible and while it is still capable enough to influence public perception. Long before the models are unable to fit the original data distribution, we would argue that immense harm would have already been caused to low-resource communities through the disproportionate erasure of their data and the environmental damage caused by training sufficiently many models to reach this point.

Thus, while Schaeffer et al. (2025) argue for a more discerning and muted rhetoric around model collapse, we are actually in agreement with their position, as they note that real tail data will be lost and that scaling laws may change with the introduction of synthetic data. They even note that "loss of diversity is a real issue, with disproportionate harms often-times born by subgroups"—pre-empting our position in this work. We may even contextualize our position as a push towards a new axis of concern around model collapse: that model collapse will happen for different communities' data at different rates and have uneven impact. For example, Schaeffer et al. (2025) note that humanity may be able to train trillions of models before we notice the onset of model collapse. This conclusion was drawn based on a premise of there being a wealth of data available. This is certainly not the case for many communities and languages. Consequently, while the discourse around model collapse may be over-cautious and pessimistic, it is certainly accurate for some low-resource communities, regardless of which definition of model collapse is ultimately favoured.

A final alternative view may consider whether the Stochastic Parrot analogy is still true, five years after the original work (Bender et al., 2021). While it is certainly true that immense progress has been made in recent years on grounded language understanding and generation (Radford et al., 2021; Rombach et al., 2022; Alayrac et al., 2022), particularly through the addition of multiple modalities, the statistical basis for the analogy remains true. LLMs base their predictions for the next token on statistical patterns which they have previously observed, which makes them susceptible to frequency and sample biases just like any other cognitive agent (Geva et al., 2025). The question then becomes: to what extend does the additional modalities allow multimodal models to ground their language generations in reality rather than the statistics of the language they encounter? This is beyond the scope of this discussion, however it appears to be of paramount importance for future

work. For now, we retain the Stochastic Parrot analogy as sufficiently relevant to provide the conceptual foundation for this work, particularly for low-resource languages and multilingual models, where clear sample biases and dominant patterns in the language data may not reflect broader societal realities (Bender et al., 2021).

# 5. Conclusions and Call to Action

In this work we have aimed to contextualize the current discussions of model collapse (Shumailov et al., 2024; Dohmatob et al., 2024a;b; Bohacek & Farid, 2023) around prior discussions on the biases and injustices perpetuated by generative models and their training (Basta et al., 2019; Bender et al., 2021). While many of the points raised here are not novel in isolation, it is the combination of perspectives we aim to contribute, as summarised in Tab.1. While some works argue that model collapse may be less severe than the current literature would lead one to believe (and importantly caution against excessive rhetoric) (Schaeffer et al., 2025), we would argue that model collapse has a higher likelihood of reaching some communities which are already marginalized. *Our position is that the degree of concern and concern should be based on the distribution, properties and quantity of a community's data*. Thus, whether one believes that model collapse is an imminent threat to our online cultures and progress in machine learning, it appears that low-resource and marginalized communities will be affected by it first and most severely. Consequently these communities should have a more prominent role in the academic discourse on the topic.

Yet we find it troubling that this has yet to occur, with most experiments remaining on theoretical model (Shumailov et al., 2024), highly controlled datasets (Wyllie et al., 2024) or high-resource settings (Wei & Tyson, 2024; Wang et al., 2025). While these highly impactful works have set the foundation for subsequent studies on model collapse, it cannot be assumed that preserving cultural sensitivity and enhancing representation of smaller communities aligns with the goals of the broader community. It is the responsibility of low-resource and marginalised communities to be proactive in maintaining the representation of their data and cultures as far as possible. Yet, we also note that the majority of people will form part of a minority group in some aspect. While axes such as race, gender and ethnicity rightly receive significant focus there are a number of different factors influencing a person's identity within their community. The homogenising effect of model collapse will affect all of these axes. Consequently, even if it is left to members within under-represented groups to understand and mitigate model collapse, model collapse still stands to be of wide-reaching concern for a large number of people. Thus, we conclude with a call to action: that the low-resource NLP and ML

communities should take the forefront in understanding and mitigating model collapse. To this end, we provide some suggestions for open questions and first steps.

The open questions we highlight revolve primarily around the differences many low-resource languages present compared to some dominant languages such as English. Characterising the consequences of these differences on model collapse appears to be of utmost importance. For example, model collapse may manifest very differently on languages which are agglutinative (such as Turkish, Korean and Swahili) (Durrant, 2013). Agglutinative languages tend to form new words by the structured composition of other morphemes. In contrast to English, which is an analytic language (with some fusional elements) (Li, 2018) and has a few highly reusable words within the vocabulary, agglutinative languages have a large vocabulary and a very long tailed distribution of word use. While LLMs typically use sophisticated tokenizers which are able to extract the morphemes from agglutination (Kudo & Richardson, 2018), this still highlights the potential for differences between languages. Even the choice of tokenizers, which have been shown to materially impact model performance on low-resource languages (Rajab, 2022), could play a key role in mitigating model collapse for these languages.

Another example of a unique property of low-resource languages is their propensity for code-switching (Olaleye et al., 2025), which occurs when words from the vocabulary of one language are used inter-changeable in another. Code-switching, in general, currently poses a technological problem for low-resource languages (Winata et al., 2023; Amol et al., 2025) and invasive words will shift the data distribution. If invasive words push other words to the tails, then parts of the original language will be forgotten. Otherwise, the invasive words will occupy the tails and be removed. In both cases the redundancy acts to increase the tail of the distribution and will impact how model collapse occurs.

These examples highlight an important point: there are many reasons why a community's data may occupy the tails of the data distribution and be at risk of model collapse. While quantity of data is a clear contributor and saliently connected to marginalisation of communities, historical discrepancies in how technology is developed can also influence the data distribution even if a huge amount of data is acquired. The ability to influence the early design of foundational technology is also highly correlated with resource access and marginalisation. Since the underlying technologies behave differently between cultures and demographics, the established shift in data distribution from model collapse (Shumailov et al., 2024; Wyllie et al., 2024) will manifest as different tangible impacts on various cultures in the real world.

Turning now towards our recommendations, we can group these into three categories: 1) data, 2) detection, 3) decreas-

ing. Regarding data, we agree with prior work on AI fairness that data curation remains an extremely valuable tool for avoiding bias and harm in LLMs (Rajab et al., 2025). The formation of datasets which have been sequestered and have guaranteed real data appears to be an imperative step (Schaeffer et al., 2025). As Schaeffer et al. (2025) note, many of these datasets already exist from the work conducted by frontier AI labs and were created naturally through the development of the early LLMs and foundation models. This remains a solution only for the languages which have already seen significant development and are likely high-resource languages. Clearly, the existing range of well-curated datasets for thousands of low-resource languages (Agić & Vulić, 2019; Siminyu & Freshia, 2020; Rajab et al., 2025) would be insufficient to preserve the authenticity of the cultures they represent if model collapse were to become a larger issue. Thus, it should be the goal of policymakers, governments and academic institutions to construct such datasets which are guaranteed to contain real data and are representative of culture. How to systematically approach such a task remains a key open question.

The second category revolves around the clear need to both detect synthetic data within datasets, so that curation can be performed more accurately, and to be able to detect model collapse when it happens. In this case, we echo the important point raised by Schaeffer et al. (2025) that clear definitions are needed and that ideally support tractable measurements for model collapse or correlates of it. How feasible it is to detect synthetic data remains to be seen, and given the challenges already shown (Doughman et al., 2025), this should not be relied upon in isolation.

Finally, strategies to decrease model collapse would aim to fix the distributional shift which results from training on synthetic data. A number of strategies may be feasible and would follow from the first two steps once we understand the problem and have data which is usable. Some places to start may be in self-supervised learning or reinforcement learning to promote robust features (Hendrycks et al., 2019) and out-of-distribution generalisation (Feng et al., 2024).

In conclusion, this work adds to the growing discussion and line of position papers (Farnadi et al., 2024; Kandpal & Raffel, 2025; Schaeffer et al., 2025) related to model collapse. While uncertainty remains around the severity and prevalence of model collapse, we argue that this will differ between communities and disproportionately affect low-resource and marginalised communities. Thus, at least some of the most vulnerable communities will face the worst case scenario of model collapse. Consequently, our position is that significantly more effort from the low-resource ML and AI fairness communities is *needed immediately* to understand how to measure and combat model collapse.

## Acknowledgments

SSM was supported by the Wallenberg AI, Autonomous Systems, and Software Program (WASP) and by OpenAI's Alignment Team, awarded by the UK AI Security Institute, through the Alignment Project. B.R. is a Canadian Institute for Advanced Research Fellow. Computations were performed using High Performance Computing infrastructure provided by the Mathematical Sciences Support unit at the University of the Witwatersrand.

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
