# OpenReview forum: "Position: the Stochastic Parrot in the Coal Mine. Model Collapse is a Threat to Low-Resource Communities"
_ICML.cc/2026/Position_Paper_Track — ICML 2026 Position Paper Track spotlight_

### Official Review · Reviewer_MmkX · 2026-03-09

**Significance:** 4
**Argument Clarity:** 3
**Rating:** 5
**Confidence:** 4

**Questions:**

- Choosing a given definition for model collapse (see section 3.3) depending on the issue being studied might introduce some bias, right? Could the authors discuss the implications of "changing the problem definition" depending on the arguments being made?
- The claim of a combined, joint discussion of various perspectives on model collapse is not fully supported my opinion. Would it be possible for the authors to explicit (even with a table or a sketch) the various connections they identified among the various risks and model collapse definitions? I think that in the current version of the paper these connections are "buried" in a (nice but) lengthy narrative.

**Alternative Views Section:**

Yes

**Compliance With Llm Reviewing Policy A Conservative:**

Affirmed.

**Discussion Potential:**

4

**Final Justification:**

As written in my message in reply to the rebuttal, I was convinced by the quality and discussion potential of this work. As such, I maintain my positive score.

**Paper Summary:**

## Summary

This work focuses on the problem of model collapse affecting LLMs, whereby models are trained using data produced by other LLMs. In other words, authors argue that the quality of data degrades, and that biases are reinforced.
The position of this paper is to state that model collapse hinders a wide adoption of AI, especially at the expenses of under-represented communities, and low-resource groups of individuals.
This work is positioned within the literature on model collapse, but emphasizes dimensions such as environmental and cultural ones.

### Details per section

- Section 1
  - This is where they root their position in some available literature. Precisely, they argue that synthetic data generation is wide-spread, and that recent LLMs are increasingly trained on such synthetic data.
  - An additional argument is that model collapse reinforces the natural propensity of LLMs to suffer from bias introduced by low-quality training data.
  - Another argument relates to energy requirements to deploy LLMs, which are prohibitive and bear consequences on the environment, which are paid globally.
  - The goal in this paper is to jointly consider these arguments, which have been discussed in isolation so far.
- Section 2
  - This is dedicated to a review of recent work on model collapse, both in theory and empirically.
  - They define model collapse as: *a decrease in the marginal value of additional (synthetic) training data and a reduction in the diversity and quality of the output distribution*.
  - Authors point at recent work indicating that even a small fraction of synthetic training data alters the exponents of the well-known scaling laws of LLMs, for the worse. Models grow a tendency to develop an inductive bias toward dominant factor of variations, caused by low-quality synthetic data.
  - Authors refer to seminal work in cognitive science, whereby the inductive biases to focus on dominant factors of variations are not necessarily negative only.
- Section 3
  - This section focuses on cost analysis.
  - Energy and investments
    - This begins with a cost analysis. Training costs energy, and this is just the tip of the iceberg: few studies include the cost of obtaining the training data, which could be as much as 10-1000 times higher than training costs.
    - Training cost estimates can be adjusted based on the percentage of synthetic data in the training set: to increase model performance when using synthetic data, dataset sizes must increase by an extremely large factor.
  - Cultural
    - Model collapse results in lack of diversity.
    - Data curation is extremely important but might not be sufficient and costly.
    - Injecting synthetic data is less costly but reinforces biases and obliterates diversity.
- Section 3.3
  - This is where authors discuss alternative views.
  - Alt 1: problem is not as severe as one might think
    - Models can stabilize and not degrade.
    - Measuring model collapse is still an open question.
    - Authors argue that such points might be somehow valid for dominant languages and cultures, not for under-represented ones.
  - Alt 2: model collapse is actually beneficial
    - This follow the "iterated learning" paradigm from cognitive sciences.
    - Model collapse could imply loss of invalid information.
  - Prior "position" papers on the topic
    - Model collapse definition is ambiguous.
    - Authors use two of the three definitions proposed in the literature.
- Section 4
  - This is where the call to action is discussed.
  - Focus on low-resource and marginalized communities, which are more affected by model collapse!

**Position:**

Yes

**Position In Title:**

Yes

**Related Work:**

3

**Strengths And Weaknesses:**

- Elegant and well constructed narrative on the problem of model collapse, and its implication especially on low-resource and marginalized communities. This is the main strength of the paper, that includes several well-selected references to position their work.
- This work considers several factors, including energy and financial costs, as well as cultural costs, and attempts (see also my weaknesses point below) to integrate those aspects under clear definitions of model collapse
- I enjoyed the connections to cognitive science, which also serve to support the alternative views section.
- The joint discussion seems a bit "buried" in the lengthy narrative of each section, and does not stand out clearly enough.

**Support:**

3

---

> ### Author Rebuttal · Authors · 2026-03-30
>
> We thank the reviewer for their consideration of our work and positive assessment. We answer each question in turn:
>
> **Question 1**: Indeed the exact definition of model collapse will impact the conclusions and risk assessment. By using multiple definitions here we aim to show that regardless of which definition you choose there is a perspective where it is plausible that low-resource communities will be disproportionately affected. Thus, for readers more concerned with the type 3 definition (decreasing improvement from more data) they should be concerned with how low-resource communities will experience slower technological advancement and that data is a hidden expense (see Kandpal et al. (2025)) which many of these communities cannot afford to obtain to compensate for model collapse. If a different reader adheres to a type 2 definition (data distribution shifts) then they would be more concerned with the erasure of culture and propagation of hegemonic viewpoints. Both readers would have cause for concern, however. The reviewer is correct though that a reader who adheres to a type 1 definition (a loss in ability of a model to learn the original data distribution) would see less immediate concern and so we do raise this as an alternative view. Our point here is to try to characterise how multiple perspectives would perceive the effect of model collapse on low-resource communities and demonstrate a need for more engagement from these communities in understanding and solving the problem. This is also why we aimed to provide a detailed alternative views section, as we aim to make our position as falsifiable as possible. To address the reviewers point, we will make it clear that our call to action includes work which aims to falsify our position as well. We are not only arguing for work showing how model collapse will harm low-resource communities disproportionately under a single definition, but rather we advocate for more work on the topic in general to understand the multiple facets of model collapse and how this affects low-resource communities. We thank the reviewer for raising this point.
>
> **Question 2**: We thank the reviewer for highlighting this weakness in our current draft and will aim to clarify the connections between the various definitions and risks we identify. To outline this here: Type 2 definitions mean that the models will scale worse under model collapse, which would exacerbate the economic risks facing low-resource communities (difficulty collecting data, loss of data sovereignty, exposure to the effects of climate change first and less access to technology, to name a few). Type 3 definitions highlight how data becomes homogenised under collapse which will exacerbate cultural impacts of LLMs on low-resource communities (the erasure of measures to reclaim words, the promotion of hegemonic viewpoints, the loss of representation on the internet). We also try to show how these impacts would interact (which is perhaps where the narrative becomes unclear). For example, in an attempt to keep up economically and overcome the costs of real data acquisition low-resource communities may still use synthetic data to a degree, which would result in the faster erasure of their data on the internet which would increase the difficulty and cost of obtaining data for these communities. For type 1 definitions, this is a longer-term issue highlighting that models themselves eventually cannot learn appropriate information from collapsing data. This definition type would conceivably still affect low-resource communities first but is less of an immediate issue and so we highlight this as a potential alternative view. We will make this structure far clearer in a subsequent draft, while still highlighting how the interaction of effects further compound the potential risks. Finally, we will include a table which outlines: 1) the different definitions of model collapse, 2) the most pressing risks which would arise from each definition, 3) some of the most relevant citations for those risks. This will serve to provide a helpful scaffold while reading our work but also may be useful in its own right as a reference for the community. We thank the reviewer again for their assistance in how we portray our message in this work.
>
> - Kandpal, Nikhil, and Colin Raffel. "Position: The Most Expensive Part of an LLM should be its Training Data." International Conference on Machine Learning. PMLR, 2025.

---

> > ### Author Rebuttal · Reviewer_MmkX · 2026-04-01
> >
> > Dear authors, thank you for reading the review and for your rebuttal.
> >
> > You answers addressed my concerns, and I will keep my positive score.

---

### Official Review · Reviewer_ounX · 2026-03-13

**Significance:** 3
**Argument Clarity:** 3
**Rating:** 4
**Confidence:** 3

**Questions:**

See Weaknesses.

**Alternative Views Section:**

Yes

**Compliance With Llm Reviewing Policy A Conservative:**

Affirmed.

**Discussion Potential:**

3

**Final Justification:**

I thank the authors for their response. I hope the authors will incorporate their response into the main paper, and I will maintain my positive score.

**Paper Summary:**

This paper argues that model collapse threatens current efforts to democratize AI, especially in low-resource and marginalized communities. The paper also explores the relationship between model collapse and cultural bias. It emphasizes that addressing the issue of model collapse should become a central topic in AI research, particularly concerning fairness and accessibility in low-resource environments.

**Position:**

Yes

**Position In Title:**

Yes

**Related Work:**

4

**Strengths And Weaknesses:**

Strengths:

The topic is of relevance and importance to the ICML community.

This paper is highly likely to inspire discussion, particularly in the fields of AI, machine learning, and technology ethics: Relevance to current AI trends, impact on marginalized communities, environmental and cultural considerations, call to action, and controversy within the AI community.

It cites related work and events appropriately. Section 2 of the paper introduces the background on model collapse and discusses related work.

Weaknesses:

The paper lacks sufficient reasoning and evidence. It needs specific cases to illustrate what constitutes a "Stochastic Parrot" and its corresponding impact.

The argument is sound. However, providing specific cases would make it clearer.

**Support:**

3

---

> ### Author Rebuttal · Authors · 2026-03-30
>
> We thank the reviewer for the constructive feedback and for the positive assessment of the paper.
>
> We agree that the current version would benefit from more concrete examples to clarify what we mean by the “Stochastic Parrot” effect and to better illustrate its real-world impact. In the revision, we will address this directly by adding specific, grounded cases (Section 3.2) showing how these effects manifest in practice, particularly for underrepresented and low-resource groups. One salient example from Wilson et al. (2025) shows how language guided diffusion models enforce racial stereotypes. While this is a multi-modal example, this propensity to repeat biased patterns in data is found in LLMs, as noted in Bender et al. (2021) - which introduced the Stochastic Parrot analogy to make this exact point. Thus, we use the “Stochastic Parrot” framing as a simplifying lens (as discussed in Section 3.3) to highlight broader risks in the evolution of generative AI for underrepresented groups. Our central concern is that these groups are likely to be the first to experience the negative effects of such systems.
>
> Extending to model collapse, our argument is that there is a compounding set of dynamics that disproportionately affect these populations. Prior work, such as Wyllie et al. (2024), shows that fairness degradation tends to appear first for minority groups when synthetic data is iteratively reused (the “fairness feedback loop”). At the same time, although there is ongoing debate about how to define model collapse (e.g., Schaeffer et al., 2025), the implications across definitions point in a similar direction: groups with less data and fewer resources are more vulnerable to degradation and face higher barriers to the effective adoption of ML models. Another salient example of how model collapse might impact society comes from Wang et al. (2025) which shows that LLMs undergoing collapse favour right-wing views. This work, however, was conducted only in English and based on US media content. It is our position that how model collapse might occur in highly different cultures, with significantly different languages and in much lower-resource settings is not being explored sufficiently in the literature given these communities' vulnerability to model collapse.
>
> Importantly, this is not just about initial harm, but about amplification. As model quality degrades faster for these groups, maintaining or recovering performance requires more high-quality data and greater computational resources. This raises the cost of parity and risks further widening existing disparities. We will incorporate these additional examples and clarify our line of reasoning in the revision to make the argument more concrete and easier to relate to tangible outcomes. We thank the reviewer again for their assistance in improving the clarity of our paper.
>
>
> - Wilson, Kyra, Sourojit Ghosh, and Aylin Caliskan. "Bias Amplification in Stable Diffusion’s Representation of Stigma Through Skin Tones and Their Homogeneity." Proceedings of the AAAI/ACM Conference on AI, Ethics, and Society. Vol. 8. No. 3. 2025.
> - Bender, Emily M., et al. "On the dangers of stochastic parrots: Can language models be too big?🦜." Proceedings of the 2021 ACM conference on fairness, accountability, and transparency. 2021.
>  - Wyllie, Sierra, Ilia Shumailov, and Nicolas Papernot. "Fairness feedback loops: training on synthetic data amplifies bias." Proceedings of the 2024 ACM Conference on Fairness, Accountability, and Transparency. 2024.
> - Schaeffer, Rylan, et al. "Position: Model collapse does not mean what you think." arXiv preprint arXiv:2503.03150 (2025).
> - Wang, Ze, et al. "Bias amplification: Large language models as increasingly biased media." Proceedings of the 14th International Joint Conference on Natural Language Processing and the 4th Conference of the Asia-Pacific Chapter of the Association for Computational Linguistics. 2025.

---

> > ### Author Rebuttal · Reviewer_ounX · 2026-04-02
> >
> > I thank the authors for their response. I have no further questions and will maintain my positive score.

---

### Official Review · Reviewer_1Vi9 · 2026-03-13

**Significance:** 3
**Argument Clarity:** 2
**Rating:** 3
**Confidence:** 4

**Questions:**

I think the paper could try to reframe itself to focus on a more specific problem needing further discussion, and my questions are about how the paper could reframe itself:

1) Could the authors further refine their position given the general observation of disproportional harm from model collapse has already been showed in the literature? A common thread was the computational/resource aspect to model collapse, and perhaps this could be the sole focus of the position paper?

2) The paper also often briefly mentions data curation and how that might help, but could the authors further elaborate on the current limitations in the data curation literature/what needs further discussion? For example there is now a plethora of methods and extensive evaluations of them (e.g., regression based and the improved reduced convex objective for the bi-level problem [4,5]), with a broad consensus that finding a good data mixture is possible by just learning a linear model/solving a convex program [5,6] (even with little target data [5]). Is there something this literature has missed?

[4] Liu, Qian, et al. "Regmix: Data mixture as regression for language model pre-training." ICLR 2025
[5] Thudi, Anvith, et al. "Mixmin: Finding data mixtures via convex minimization." ICML 2025
[6] Chen, Mayee F., et al. "Olmix: A Framework for Data Mixing Throughout LM Development." arXiv preprint arXiv:2602.12237 (2026).

**Alternative Views Section:**

Yes

**Compliance With Llm Reviewing Policy A Conservative:**

Affirmed.

**Discussion Potential:**

2

**Final Justification:**

The authors have elaborated on what the past literature is specifically missing, and so I raise my score to a 3. I keep my score at a 3, as I think making this clearer in the paper would require some significant rewrites, and perhaps also changing the wording of the position itself.

**Paper Summary:**

The paper discusses how model collapse could disproportionately affect low-resource/minority groups in a dataset, causing more harm to them than to other groups. They make this argument by discussing how model collapse results in a loss of the tails, and that it is likely these group fall in these tails and hence would foreseeably be lost first. They also discuss how the need for compute to mitigate model collapse might also push low-resource communities from being able to have performant models, or push to using more synthetic data to further representation (which might then cause more harm). They discuss alternative views, such as model collapse would only occur to parts of the distribution which are not stable/reoccuring over time.

**Position:**

Yes

**Position In Title:**

Yes

**Related Work:**

2

**Strengths And Weaknesses:**

Strengths:

1) The topic that model collapse has disproportionate impact is relevant and important to the comminity

Weaknesses:

I found this paper completely missed or misrepresented existing results. In particular, they cite Wylie et al., but do not mention how in that paper there are already experiments explicitly showing many of the "possible" issues this paper discusses. For example in that paper they had experiments showing generative models undergoing model collapse first stopped producing minority groups, and they also showed how predictive models quickly became less fair for those minority groups (as measured by several metrics). This work also has several follow-up works, all of which are missing in this paper (for example [1] showed how model collapse LLM amplifies right-wing views, and [2] shows specific bias in skin tones). Considering the whole position is model collapse "could" cause disproportionate harm, and this has actually already been extensively showed and discussed in the literature, I found the position while interesting lacking enough background to meaningfully push what is already known in the literature.

Moreover, specific discussion points also lacked references/context to existing literature. For example, the paper discusses how synthetic data could be used to supplement minority data groups, and [3] already discusses the issues of this.


[1] Wang, Ze, et al. "Bias amplification: Large language models as increasingly biased media." Proceedings of the 14th International Joint Conference on Natural Language Processing and the 4th Conference of the Asia-Pacific Chapter of the Association for Computational Linguistics. 2025.
[2] Wilson, Kyra, Sourojit Ghosh, and Aylin Caliskan. "Bias Amplification in Stable Diffusion’s Representation of Stigma Through Skin Tones and Their Homogeneity." Proceedings of the AAAI/ACM Conference on AI, Ethics, and Society. Vol. 8. No. 3. 2025.
[3] Whitney, Cedric Deslandes, and Justin Norman. "Real risks of fake data: Synthetic data, diversity-washing and consent circumvention." Proceedings of the 2024 ACM conference on fairness, accountability, and transparency. 2024.

**Support:**

2

---

> ### Author Rebuttal · Authors · 2026-03-31
>
> We thank the reviewer for their review and for pointing us toward these references. Below, we comment on our position, the missing literature, and the data curation questions.
>
> **Clarifying our position**: The reviewer is right that the harms around model collapse and synthetic data are already appearing in the literature. Our contribution, which we will articulate more clearly, is to observe that if disproportionate harm is already measurable in well-resourced, well-studied settings with relatively clean datasets, then it is plausible that the situation for low-resource communities is worse. Critically, we bring together these emerging findings while discussing their potential implications for low-resource communities. Narrowing the scope of our paper would undermine this contribution. It is key to note that even low-resource languages in isolation present compounding issues that are absent from current empirical studies on model collapse: existing misrepresentation in data, agglutinative morphology, metaphorical phrasing, unreliable tokenization, widespread code-switching and reader-responsible discourse norms. All of these examples differ fundamentally from the high-resource and typically English texts on which most current collapse experiments have been conducted for language. More broadly, none of the existing empirical work (valuable as it is) addresses these conditions, and results from English, CelebA, coloured MNIST and US media alone cannot be assumed to transfer to low-resource communities or cultures. This gap is precisely what our call to action targets. While the proposed citations are relevant and we will gladly include them, none of them undermine our call to action or the main points of our position
>
> **Regarding Wang et al. (2025)**: This is an exceptional example showing how political discourse can be shaped by collapsing models. It is, however, based solely on English data from US media outlets. The extreme outcome shown in that paper, along with the high-resource nature of the study, demonstrates our point perfectly. We are calling for more studies on this topic for low-resource communities because of the stark outcomes we are seeing from model collapse in high-resource communities. We will certainly cite this work in a revised draft to motivate our position.
>
> **Wilson et al. (2025) & Whitney et al. (2024)**: Wilson et al (2025) is not about model collapse and only mentions this as future work. However, the clear discrimination they present from diffusion models on marginalised skintones is another clear motivator for our position here. Diffusion models are discriminating on skintone as it is; we strongly believe that if model collapse is going to worsen this effect, then much more research is needed. Similarly, Whitney et al (2024) do not mention model collapse but do raise other ethical concerns about synthetic data use (leading to overly high confidence in classifier models, circumventing consent, and how synthetic data can be used to deliberately consolidate power around majorities).
>
> **Wyllie et al. (2024)**: As the reviewer notes we do already cite Wyllie et al. (2024) as it is a very relevant prior work which does focus on model collapse. However, the datasets considered by Wyllie et al. (2024) are: coloredMNIST and coloredSVHN as toy datasets, and FairFace and CelebA as the more natural datasets. All of these are image datasets and relatively small. Thus, while Wyllie et al. (2024) is an exceptional work it remains fairly theoretical and does not consider datasets taken directly from low-resource communities that are grounded in the cultures of these communities.
>
> **Regarding Chen et al. (2026) and the Limitations in data curation**: *Chen et al. (2026) appeared on ArXiv after the ICML submission deadline*, which is why we could not have included it originally. That said, it aligns closely with our concerns and we agree that these citations need to be addressed more directly. Specifically, Chen et al. (2026) note in their abstract that: *“While existing mixing methods show promise, they fall short when applied during real-world LM development”*. Moreover, these methods generally assume access to enough high-quality target data to guide the optimisation process. For low-resource languages, that assumption often doesn’t hold. We will incorporate these points to emphasise that (even where there is theoretical agreement), practical, robust solutions for low-resource contexts are still lacking, which would include Liu et al [4] and Thudi et al [5] as well. Thus, even the suggested citations seem to agree that at the time of the ICML submission deadline, finding the optimal data mixture in real-world applications is an open problem. We will highlight the potential of Chen et al. (2026) in this paper as it may prove to be very impactful for the field.

---

> > ### Author Rebuttal · Reviewer_1Vi9 · 2026-04-03
> >
> > My main concern was that given many of the "potential" harms mentioned in the paper are known in various contexts, the focus of discussion should then be around what such existing literature is missing. The authors state none of these are "low-resource" settings, but I find this a bit ill-posed. One could argue a minority group in an image dataset is a low-resource group.
> >
> > Moreover on the dataset curation, I believe in Thudi et al., they were able to find useful curations of data for target datasets with 100 or fewer positive samples (e.g., many of the bio assays on PubChem). The authors state they did not provide solutions for low resource settings, but I find this makes the notion of low-resource more confusing. I agree Chen et al came after the current paper, but I mentioned that paper to further show the trends from earlier data mixing works have been reproducable. In particular Chen et al themselves missed some other literature (e.g., Thudi et al.,) but came to the same trends on a linear model was sufficient for data mixing. Given other work already also shows this independent of them, I understand the consensus in the literature to be data mixing is generally an "easy" learning problem (a linear model).
> >
> > Given I am now actually less sure of the contributions of this discussion beyond what existing literature already points out, I am reducing my score to a 1.

---

### Official Review · Reviewer_t1ar · 2026-03-13

**Significance:** 3
**Argument Clarity:** 3
**Rating:** 5
**Confidence:** 2

**Questions:**

1. Is it possible to successfully combat the discussed effects of the model collapse without introducing significant economic changes first?

**Alternative Views Section:**

Yes

**Compliance With Llm Reviewing Policy A Conservative:**

Affirmed.

**Discussion Potential:**

3

**Final Justification:**

In light of the discussion between the authors and Reviewer 1Vi9, I decrease my confidence, since I am not quite familiar with the related works.

**Paper Summary:**

This paper systematically examines the role of model collapse (a result of generative models being trained on the outputs of prior models) in reinforcing challenges faced by small or marginalized communities, such as biases, underrepresentation, and limited resources. The authors argue that model collapse poses a significant threat to these communities. They conclude with a call to action, urging increased research and effort to study and mitigate the causes and effects of model collapse.

**Position:**

Yes

**Position In Title:**

Yes

**Related Work:**

4

**Strengths And Weaknesses:**

**Strengths:**

- The paper is well-written and easy to follow.
- The evidence base is rather rich: many works on specific issues (e.g., the difficulty of data scraping, environmental impact caused by AI, transfer learning, high cost of training, and other) are employed to support the authors' position.
- The "Alternative Views" section does not rely on straw‑man arguments, and provides some interesting opinions (for example, the notion that the collapsed state converges to the language most grounded in reality). Therefore, this paper would probably spark a well-rounded scientific discussion.

**Weaknesses:**

1. Although the authors do a good work grounding their claims in the existing literature, the article would benefit from a more direct set of evidence. Below are some examples and suggestions.
   - Some experiments that demonstrate, in a multilingual scenario, how collapsing models first start to forget the languages at the tails of the *combined* distribution could be performed to substantiate the claim on lines 223-227.
   - Perhaps, a small model that tackles both the AI and economic aspects of the problem, and corresponding numerical experiments could be leveraged to better illustrate the connection between the model collapse and economic development. This approach could also be employed to add some quantitative analysis to the claim on lines 375-376 ("model collapse will happen for different communities' data at different rates and have uneven impact")
1. In light of the previous weakness, it is still hard to conclude whether the model collapse indeed accelerates the deepening of the gap between marginalized and non-marginalized communities, or rather it serves merely as a small part of an already existing trend, thus not accelerating it in any significant way. To properly address this question, one has to somehow quantify the discussed influence of model collapse.
1. I find that the fundamental question, whether preserving cultural authenticity and enhancing representation of small communities aligns with the goals of larger communities, is unaddressed in the article. For the sake of completeness, and to ensure better understanding of the work among a wider audience, I kindly ask the authors to include a corresponding discussion in the work.

**Support:**

3

---

> ### Author Rebuttal · Authors · 2026-03-30
>
> We thank the reviewer for their detailed consideration of our work and overall positive evaluation. We will discuss each of the noted weaknesses in turn.
>
> **Weakness 1**: Figure 1 as it stands is meant to lend credence to our position that low-resource languages will be forgotten first. The distribution shown in that figure is the combined distribution from GPT-2, just coloured by language. However we acknowledge that the collapse is not shown directly. We will endeavour to add this result in a subsequent draft. We thank the reviewer for this suggestion.
>
> **Weakness 2**: This is a significantly bigger question, and indeed a weakness of the field when discussing model collapse (as noted by Schaeffer et al. (2025)). Defining and quantifying model collapse generally is a very important open question and our position is that there is a need for marginalised and low-resource communities to be involved in forming those definitions. For example, how would one even measure the loss of a culture? The perplexity of the culture’s data currently plays a surrogate role for this measure, but more work is needed to determine if this is sufficient. Additionally, our strategy here has been to provide sufficient evidence to support our position from multiple perspectives while also presenting concrete claims to remain falsifiable. We have endeavoured to 1) show how multiple perspectives and measures of the impact of collapse would all lead to the concern that low-resource communities could be disproportionately affected; 2) presented a fair and balanced alternative views sections to also help or direct future work which aims to falsify our position. Indeed there is room to believe that it will not: consider Dohmatob et al. (2024) (Figure 3) for example. Low-resource communities may be on a portion of the neural scaling law trend which does not plateau while high-resource communities are. However, collapse occurs faster as a greater proportion of a data distribution becomes synthetic (also shown in Dohmatob et al. (2024) (Figure 3)) and it will be far easier to overwhelm the real data on low-resource settings by definition. Our position is not that we know how model collapse will affect low-resource communities. Our position is that low-resource communities should be at the forefront of the research to find out. We do acknowledge that the point on whether model collapse is accelerating an existing problem is a core distinction which we have not made explicit enough in the current draft. So we thank the reviewer for highlighting this and will gladly revise our Alternative Views and Call to Action sections to raise this explicitly as one of the questions needing immediate research action.
>
> **Weakness 3**: We thank the reviewer for highlighting this point too and we will add the requested discussion around this. It is very likely that the answer to this question will vary across individuals. Low-resource ML communities across the world must take responsibility for their own preservation as a result and cannot rely on 1) the high-resource communities to conduct research for low-research communities; 2) for high-resource communities research on themselves to inherently transfer to low resource communities. This is why it is so concerning to see that low-resource communities are relatively absent from the current discussion on model collapse, motivating our call to action. Why is this relevant to the ICML community, then? The vast majority of the data on the internet is in English. If low-resource languages or communities' data is lost or corrupted this is a point of international concern and will affect a significant proportion of the ICML readership and the communities they come from. Examples such as Wilson et al. (2025) which show that diffusion models (without even considering model collapse) promote stereotypes around skin colour supports this. Finally, very many people are members of a low-resource or marginalised community in some facet, be it through race, gender, sexuality, affiliation or some other orientation. It seems unlikely that the majority of people could let low-resource communities be eroded and remain unaffected. Once again, we think this could be more explicit in our paper and thank the reviewer for the constructive review.
>
> **Question 1**: It is plausible to believe that, with sufficient research attention, the effects of model collapse could be mitigated through algorithmic means. We are hopeful that the mechanism which enforces expressivity in human language can be understood and utilised in a manner which protects all communities from suffering the effects of model collapse. However, communities will still need to be able to develop and deploy these solutions and any delay in availability of the solutions will also promote inequality. Thus, low-resource communities are not in a position to wait for a solution either.

---

> > ### Author Rebuttal · Reviewer_t1ar · 2026-04-05
> >
> > I thank the authors for their thorough response to my concerns.
> >
> > Given that the total rebuttal length is limited and the conference does not allow intermediate revisions to be uploaded, I am mostly satisfied with the response. However, I still think that the paper would benefit greatly from extensive research on W1 and W2. Therefore, I cannot advocate for a higher score. On the other hand, I agree with the authors that such research would be rather expensive and not core to the paper's contribution. Thus, I believe that this paper should be accepted nonetheless, given that all rebuttal amendments and clarifications are introduced in the text.
> >
> > I also acknowledge the discussion with Reviewer 1Vi9 regarding novelty. Unfortunately, my knowledge of this field is quite limited, so I cannot fully assess the arguments from both sides. Nevertheless, I agree with Reviewer 1Vi9 that the paper's contribution should be articulated more clearly, and that further clarifications are needed regarding the differences between, for example, minorities and low-resource communities.

---

### Decision · Program_Chairs · 2026-04-30

**Decision:**

Accept (spotlight)

**Comment:**

This paper agues that model collapse is of particular importance to low-resource languages/communities.  It causes models to focus on the middle of a distribution rather than the tail.

A major concern was that the paper failed to differentiate itself well from previous work, and one reviewer worried that the authors may actually misrepresent previous work in order to over-emphasize the differences.  There was robust reviewer discussion where this was defended against, as were the lack-of-novelty claims - in particular this paper blends low resource concerns with bias concerns which may be more novel.

In general the authors did a nice job of handling the critiques raised and I believe this paper raises interesting and important points.